

# Balance between driving stress and basal drag results in linearity between driving stress and basal yield stress in Antarctica's Siple Coast Ice Streams

Jan Wohland [1,2], Torsten Albrecht [1], and Anders Levermann [1,2,3]

[1]Potsdam Institute for Climate Impact Research (PIK), Potsdam, Germany
[2]Institute of Physics, University of Potsdam, Potsdam, Germany
[3]Lamont-Doherty Earth Observatory, Columbia University, New York, USA

*Correspondence to:* A. Levermann (levermann@pik-potsdam.de)

**Abstract.** Ice streams are distinct, fast-flowing regimes within ice sheets that exhibit fundamentally different characteristics as compared to the slow-moving inner parts of the ice sheets. While along-flow surface profiles of ice sheets are typically convex, some ice streams show linearly sloping or even concave surface profiles. We use observational data of the Siple Coast in Antarctica to

inversely calculate membrane stresses, driving stresses and basal yield stresses based on the Shallow Shelf Approximation. Herein we assume that these marine-based ice streams are isothermal and in neglecting vertical shear we assume that their flow is dominated by sliding. We find that in the Siple Coast ice streams the membrane stresses are negligible and the driving stress balances the basal drag. It follows directly that in the Coulomb limit (i.e. basal drag independent of velocity) the

driving stress is linear in the basal yield stress. In addition, we find that the ice topography and the basal conditions developed such that the driving stress is linear in the basal yield stress regardless of the choice of the pseudo plastic exponent in the basal drag parameterization.

## 1 Introduction

The surface elevation of the large grounded ice sheets of Antarctica and Greenland usually shows

convex curvature in the sense that its downward surface slope increases towards the coast. In the case of shear-dominated flow, this shape is supported by one-dimensional analytical steady state solutions of the Shallow Ice Approximation (SIA; Vialov, 1958). However, observations show that along-flow elevation profiles, e.g. in the narrow ice streams of the Siple Coast, West Antarctica, can be linear or even slightly concave (see Fig. 1). While the surface slope in the upper part of Ice Stream A is steeper

than in its lower part, Bindschadler Ice Stream has a nearly constantly declining surface slope over



its entire length. Mercer, Whillans and MacAyeal Ice Streams lie between this range of concave and linear shapes (Fretwell et al., 2013).

Linearly sloping surfaces seem to imply a paradoxon: On the one hand the driving stress declines towards the coast because the ice thickness decreases while the surface slope remains constant. Such 25 a decrease in the driving stress would generally lead to decreasing velocities. On the other hand mass continuity requires velocities to increase downstream for decreasing ice thickness. A common explanation of this phenomenon is based on basal friction that diminishes along the flow (Whillans and van der Veen, 1993; Bindschadler and Choi, 2007). Other mechanisms, such as stream widening or strain heating at shear margins, also provide possible explanations (Suckale et al., 2014).

Basal processes underneath ice streams are among the most uncertain boundary conditions in ice flow modeling. However, research on hydrology has lead to a significant improvement of knowledge about subglacial processes within the last couple of years (Siegert et al., 2016a). These advances include better understanding of the creation and dynamical influence of subglacial lakes (Sergienko and Hulbe, 2011; Bell et al., 2007; Siegert et al., 2016b), potential positive feedback loops between 35 rapid ice flow and availability of subglacial water (Bindschadler and Choi, 2007) or tidally modulated stick-slip motion of the Whillans Ice Stream (Wiens et al., 2008; Goldberg et al., 2014). In summaray, these findings reveal a complex and far-reaching interaction of ice, bedrock, hydrology and ocean.

In this complex system, hydrology reacts on the shortest timescales (Siegert et al., 2016b) and might thus play a crucial role in decadal to centennial alterations. Furthermore Wright et al. (2008) 40 reveal a very strong dependency of hydrological flows on ice surface geometry. They report that small surface elevation changes of the order of 10 meters are sufficient to inverse the direction of subglacial hydrological flows from some of the largest subglacial lakes in Antarctica and can thus alter heat fluxes and water availability substantially (Wright et al., 2008).

Moreover, ice streams can migrate and some are reported to have switched between active and 45 passive mode in the past (Gades et al., 2000; Winter et al., 2015). Nowadays, for example Kamb Ice Stream is dormant. This stream formation and decay can be reproduced using a temperature dependent sliding law in a conceptual radially symmetric setup (Brinkerhoff and Johnson, 2015). Moreover, the spatial configuration of ice streams can be linked to subglacial hydrology. Kyrke-Smith et al. (2015) use a coupled model, which captures ice dynamics and subglacial hydrology, to 50 assess spatial aspects of ice stream formation and find a natural length scale for stream separation.

Many glaciers lie on a bed composed of till, a shallow layer of sedimentary material. Different from glaciers that rest directly on hard bedrock, basal motion of the ice can either be caused by deformation in this boundary till layer (similar to a conveyor belt) or by sliding of the ice over the till. The contribution of till deformation, i.e. vertical shear within the till layer, to the basal ice velocities 55 can be large or even dominant, for example in Breidamerkurjökull (Iceland) till deformation accounts for 88% of basal ice velocities (Van der Veen, 2013) while the remainder is due to sliding. Similary, the high velocities of Whillans Ice Stream are largely due to till deformation below the ice according



to experimental data taken at the Upstream B Camp (Alley et al., 1986). In contrast, more than 80% of basal motion in Bindschadler Ice Stream is caused by sliding of the ice column (Cuffey and

Paterson, 2010). These different regimes (domination of shear in the till layer versus domination of sliding of the ice column over the till layer) indicate that basal conditions vary strongly within the Siple Coast and hint to the complexity of the subglacial system.

Taking note of this multitude of potentially relevant processes and scales, our analysis aims to understand fundamental aspects of the basal conditions in the ice streams of the Siple Coast. We

therefore follow a reduced model approach which does not capture many of the processes explicitly but reveals simple relationships between the main determinants of ice motion based on data inversion.

The Shallow Ice Approximation (SIA) and the Shallow Shelf Approximation (SSA) are commonly applied simplifications of the Stokes equation for ice sheets and ice shelves with much larger

horizontal than vertical extent (Greve and Blatter, 2009). In some specific configurations certain stress terms within these equations dominate, while others can be neglected. For example, in the derivation of a semi-analytic solution of the SSA using a boundary layer theory, Schoof (2007) argues that the membrane stress (which is called 'longitudinal stress' therein) is negligible in the interior of the ice sheet since it is only a higher order correction to the force balance (Schoof, personal

communication, 2016). Schoof's analytical argument is found to hold in the Siple Coast Streams as well in this study, where we first approach basal processes from an inverse modeling perspective to assess the relative contribution of different stresses in the Siple Coast ice streams. Secondly, we perform linear regression analyses to test for linear dependence between the basal yield stress and the driving stress in these ice streams.



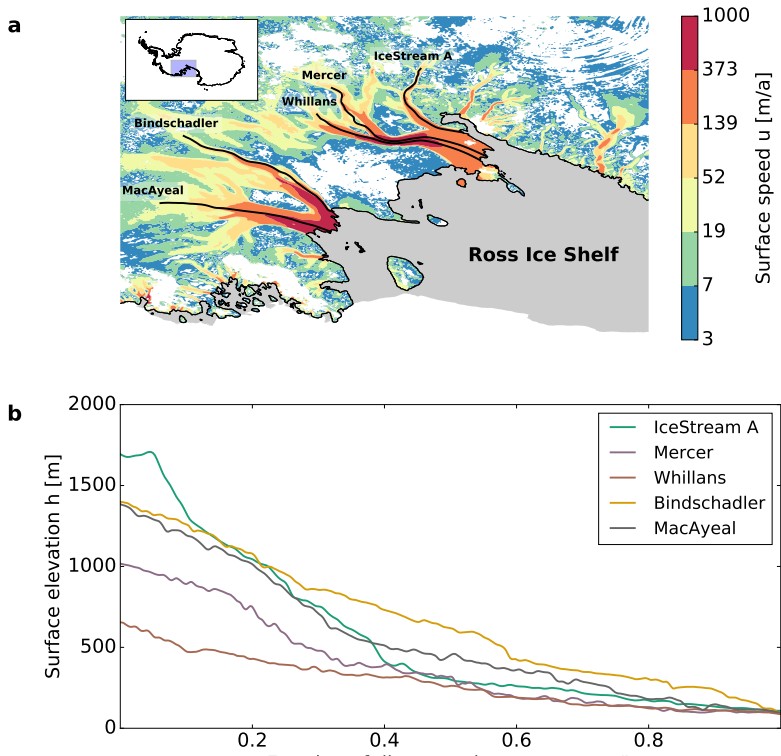

Figure 1: (**a**) Surface speeds in the Siple Coast (Rignot et al., 2011). Black lines indicate flowlines along which data is taken for the regression analysis. White areas denote speeds smaller than $3\,\mathrm{m/a}$ (over land) or open ocean. Floating ice is displayed using gray shading. The onset of the streams is chosen manually and they follow the surface flow down to the grounding line. (**b**) Surface elevation profiles, taken from the Bedmap2 dataset (Fretwell et al., 2013) along the flowlines indicated in (a), are nearly linear with a slight tendency towards concavity, or strongly concave in the case of Ice Stream A. The profiles are normalized for different ice-stream lengths $L$ between $350\,\mathrm{km}$ and $530\,\mathrm{km}$.

## 2 Underlying assumptions and theory


There are a number of assumptions that we have made in the analysis of the data and all results are subject to the constraints that arise through theses assumptions. First, we consider the ice streams to be one-dimensional structures and assume that their dynamical behavior is sufficiently represented by the SSA. Hence we assume vertical shear in the ice column to be negligible and do not solve the



SIA. Furthermore, we do not include a till model. As a consequence, ice motion is considered to arise solely from sliding along the ground, which is represented here as by the SSA velocity component $u$ (Bueler and Brown, 2009). The corresponding SSA equation in one dimension is

$$2E_{\mathrm{SSA}}^{-\frac{1}{n}}\left[HA^{-\frac{1}{n}}\left(u_x\right)^{\frac{1}{n}}\right]_x + \tau_b = \rho_i gHh_x, \tag{1}$$

where $H$ denotes ice thickness, $h$ denotes surface elevation, $\tau_b$ is basal drag, $n = 3$ is Glen's ex-
ponent, the SSA enhancement factor $E_{\mathrm{SSA}}$ is chosen as $0.6$ is and $g$ is gravitational acceleration (Winkelmann et al., 2011). We set the ice density $\rho_i$ to $900\mathrm{kg/m}^3$, in accordance with the value used in the MISMIP experiment (Schoof et al., 2009). The index $x$ denotes derivatives with respect to the spatial coordinate (e.g. $\frac{\partial h}{\partial x} = h_x$). We use a constant ice softness $A = 10^{-25}\mathrm{Pa}^{-3}\mathrm{s}^{-1}$ (isother-mal) and a Weertman-type power law parameterization of the basal drag $\tau_b$. Restricted to the positive
half space $x \geq 0$ and positive velocities this reads

$$\tau_b = -\tau_c \left(\frac{u}{u_t}\right)^q, \tag{2}$$

where $\tau_c$ denotes the basal yield stress, $u$ is ice velocity, $u_t = 1\ \mathrm{m/s}$ is a fixed velocity scale and $q$ is called pseudo plastic exponent.

Accumulation $a$ is assumed to be constant over the considered domain (in accordance with data
of a regional climate model (RACMO2) (van den Broeke et al., 2006)) and we choose $a = 0.5\,\mathrm{m/a}$ which has the same order of magnitude as in MISMIP where $a = 0.3\,\mathrm{m/a}$ as default (Pattyn et al., 2012). In a sensitivity test within the MISMIP framework $a = 0.5\,\mathrm{m/a}$ was also explored. Moreover, we introduce an onset flux $Q_0$, which accounts for the flux from the upstream catchment area into the ice streams. Hence the mass continuity equation in steady state yields

$$u = \frac{Q_0 + \int_0^x a(x')dx'}{H} = \frac{Q_0 + ax}{H}, \tag{3}$$

where we choose a coordinate system such that $x = 0$ denotes the beginning of the domain in which we consider the flowline assumptions to be valid. The onset flux is estimated as $Q_0 = u(x = 0)H(x = 0)$, where velocity $u$ and ice thickness $H$ are taken from observational data sets ((Rignot et al., 2011) and (Fretwell et al., 2013), respectively). Since the catchment area upstream is far big-
ger than the area of the ice stream, the onset flux $Q_0$ dominates over the additional fluxes due to precipitation along the ice stream $ax$.

The Shallow Shelf Approximation (Eq. 1) is a non-local elliptical differential equation that usually needs to be solved numerically. However, if its solution is already known (e.g. from observations) calculations become straightforward in the flowline case. Our general strategy is to use the thickness
$H$ and surface elevation $h$ fields from the Bedmap2 dataset (Fretwell et al., 2013) and replace veloc-ities following continuity Eq. 3. Based on this, we derive expressions for the different stresses that depend on ice sheet topography. Following this strategy, we insert Eq. 2 into Eq. 1 and solve for the





basal yield stress which leads to

$$\tau_c = \left(\frac{u_t}{u}\right)^q \cdot \left\{ 2(AE_{\text{SSA}})^{-\frac{1}{n}} \left[ H v_x^{\frac{1}{n}} \right]_x - \rho_i g H h_x \right\}, \tag{4}$$

where we used that the ice softness $A$ is assumed to be constant. We rewrite Eq. 4 schematically as

$$\tau_c = \alpha \cdot \{\tau_m + \tau_d\} \tag{5}$$

and refer to $\tau_c$, $\tau_m$, $\tau_d$ as (basal) yield stress, membrane stress and driving stress, respectively. Since the velocity $u$ can be rewritten in terms of the ice geometry (Eq. 3), the factor $\alpha$ is given as

$$\alpha = \left(\frac{u_t H}{ax + Q_0}\right)^q. \tag{6}$$

The driving stress

$$\tau_d = -\rho_i g H h_x \tag{7}$$

is known because the ice topography and hence the spatial derivative of $h$ is known. The membrane stress can be rewritten using Eq. 3 and some calculus (see Appendix) as


$$\frac{\tau_m}{2(AE_{\text{SSA}})^{-\frac{1}{n}}} = H_x \left( \frac{a}{H} - \frac{ax + Q_0}{H^2} H_x \right)^{\frac{1}{n}} + \frac{H}{n} \left( \frac{a}{H} - \frac{ax + Q_0}{H^2} H_x \right)^{\frac{1}{n} - 1} \cdot$$
$$\cdot \left\{ -2\frac{a}{H^2} H_x - (ax + Q_0) \left( \frac{1}{H^2} H_{x^2} - \frac{2}{H^3} (H_x)^2 \right) \right\}. \tag{8}$$

Note that Eq. 8 is undefined (i.e. $\tau_m$ is a complex number) if

$$aH - (ax + Q_0)H_x < 0. \tag{9}$$

Since all elements except for $H_x$ are non-negative by definition, this criterion can only be met if the thickness increases downslope ($H_x > 0$).

The above set of equations allows for a computation of the basal yield stress if the ice stream geometry ($H$ and $h$) is given. Since the solution of the partial differential equation is already known, $\tau_c$ becomes a local function of the ice sheet geometry in this particular case where the geometry

is given a priori. That is in stark contrast to typical forward modeling approaches that include an evolution of the ice sheet geometry, where the solution of the SSA requires integration over the entire modeling domain.



### 3 Stresses based on Bedmap2 dataset in Siple Coast ice streams

We use ice thickness and surface elevation of the Bedmap2 dataset provided on a $1\,\mathrm{km}$-grid to invert

for the basal yield stress. Uncertainties in surface elevation and ice thickness in the Siple Coast area are estimated following Fretwell et al. (2013) to be $30\mathrm{m}$ and $80\mathrm{m}$, respectively. We perform a forward running mean smoothing of the data over 50 grid points (accounting for roughly 10% to 15% of ice stream length) which we consider justified because the derivatives of ice thickness further amplify the substantial noise in the data. This smoothing avoids unrealistic solutions of the inversion

over wide areas of the streams.

#### 3.1 Balance between driving stress and basal drag

The ratio of driving stress and membrane stresses based on smoothed data along the flowlines (depicted in Fig. 1a) is displayed in Fig. 2a. There is strong local variation (up to three orders of magnitude) of the ratio despite the low-pass smoothing of the data. However, we find that the membrane

stress is clearly smaller than the driving stress over major parts of the ice stream (excluding the vicinity of the grounding line).

It follows that Eq. (5) can be reasonably accurate simplified to

$$\tau_c = -\alpha \cdot \tau_d, \tag{10}$$

which means that the basal yield stress and the driving stress in these regimes are coupled via the

factor $\alpha$ of Eq. 6. In the Coulomb limit ($q \to 0$) the factor $\alpha$ becomes a constant, thus leading to proportionality of yield stress and driving stress. This limit is discussed to be a valid approximation of basal drag in the transition zone (e.g. (Leguy et al., 2014)).

The dominating driving stress has two sources of uncertainty and we calculate the propagated errors following standard gaussian error propagation which yields

$$u(<\tau_d>_i) = <\tau_d>_i \sqrt{u_H^2 \sum_{j=i}^{i+N} H_j^{-2} + \left(\frac{2u_h}{h_{i+N+1} - h_i}\right)^2}, \tag{11}$$

where $<\tau_d>_i = \frac{1}{N} \sum_i^{i+N} \tau_{di}$ denotes the running mean at point $x_i$, $u_H = 80\mathrm{m}$ is the uncertainty of ice thickness data and $u_h = 30\mathrm{m}$ denotes the surface elevation uncertainty (values taken from (Fretwell et al., 2013)). The driving stress and its uncertainty are shown in the middle panel of Fig. 2b. One part of the surface elevation uncertainty, namely the systematic error (also offset or

bias), does not affect the uncertainty of the driving stress because offsets are eradicated by the first derivative. Although the shown uncertainties can thus be seen as conservative estimates, we find the driving stress signal to be stronger than the uncertainty range.

In Whillans Ice Stream, the membrane stress is undefined around $x = 100\,\mathrm{km}$. In fact we find a solution for data subjected to longer running mean intervals, but then we would loose more spatial

information.





The data shows an overall decrease of $\alpha^{\frac{1}{q}}$ (Eq. (6)) along the flowline for all streams (see Fig. 2c). This tendency is to be expected as the ice thickness generally decreases in downstream direction, resulting in a decrease in the numerator while the denominator increases. However, as the ice thickness does not decrease monotonically but can even increase on smaller spatial scales, local increases in $\alpha^{\frac{1}{q}}$ are possible. The magnitude of the decrease does vary strongly from stream to stream. While Ice Stream A shows a pronounced maximum at the stream onset, Whillans and Mercer reveal almost no changes in $\alpha^{\frac{1}{q}}$ along the flow. Note that in all cases where $0 < q < 1$, which includes the common choice $q = 0.33$, the change in $\alpha$ is smaller than the change in $\alpha^{\frac{1}{q}}$.

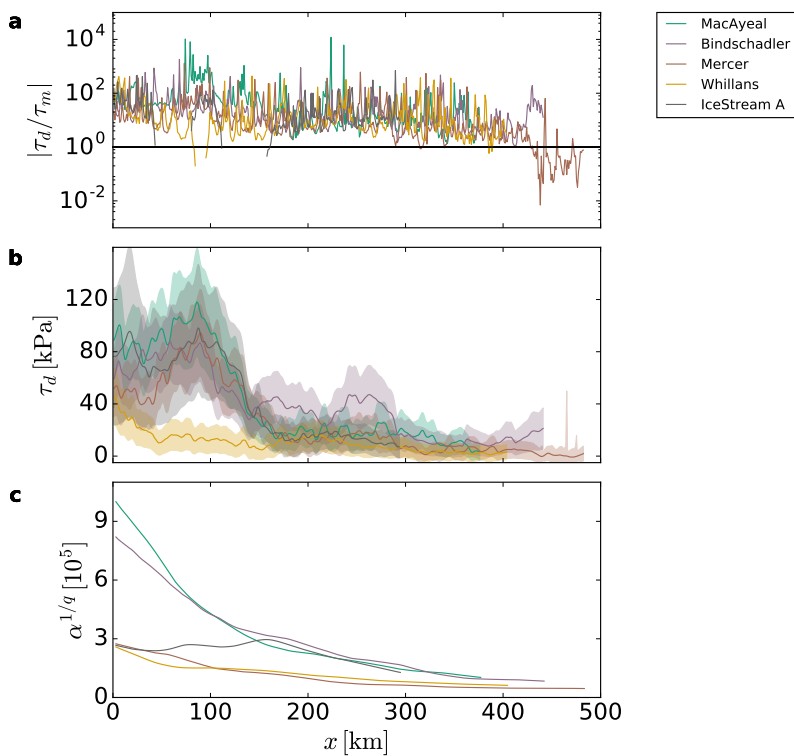

Figure 2: **(a)** Ratio of calculated driving stress and membrane stress, **(b)** driving stress and corresponding uncertainty range and **(c)** factor $\alpha^{\frac{1}{q}}$ along the flowlines. Note that the lower panel shows $\alpha^{\frac{1}{q}}$, which is independent of $q$.





### 3.2 Linear relation between driving stress and basal yield stress

We have shown that the membrane stress is clearly negligible in most regions of the Siple Coast ice streams and motivated that, as a consequence, we expect a linear relationship between the driving stress and the basal yield stress in the Coulomb limit. Now we evaluate the interdependence of the driving and yield stress for different choices of the pseudo plastic exponent $q$. Note that the full equation (5) is solved for the yield stress, i.e. the membrane stress is computed as well, although we

already argued that it generally plays a minor role. Figure 3 shows scatterplots of the computed yield stress as a function of the driving stress.

In order to quantify the linearity, a correlation analysis based on a linear regression is performed (see Table 1). The underlying fit function is $\tau_c = \tau_{c,0} + a\,\tau_d$, where $\tau_{c,0}$ and $a$ are constants, meaning that we allow offsets of the yield stress. We find that there is high correlation, which we define

here as correlation coefficient $R^2 \geq 0.9$, for the ice streams Bindschadler, Mercer, MacAyeal and Ice Stream A for $q \leq \frac{1}{3}$. Whillans Ice Stream shows a weaker correlation which can probably be attributed to a remaining singularity of the membrane stress which is still not eradicated by the applied smoothing.

Linearity between yield stress and driving stress weakens for $q = 1$ since the factor $\alpha$ in Eq. (10)

becomes more relevant. In MacAyeal, Bindschadler and Mercer Ice Stream the data deviates from linearity at high values of the driving stress. This data corresponds to the uppermost fraction of the streams (see triangles in Fig. 3). This behaviour might be due to an altered flow regime at the onset of the streams, that is not adequately represented by SSA stress balance (Note that the location of stream onset has been chosen manually and not in terms of a defined threshold). Notwithstanding,

there is fairly good correlation of the driving stress and the yield stress ($R^2 \geq 0.8$) despite the meandering.

Generally the correlation is lower between yield stress and the components of the driving stress, i.e. surface slope ($R^2 \geq 0.73$) and ice thickness ($R^2 \geq 0.23$). In the case of a linearly sloping surface

($\frac{dh}{dx}$ =const), similar to the one of Bindschadler Ice Stream, the driving stress becomes proportional to the ice thickness $H$ and we thus find high correlations between them. This proportionality corresponds to a simple form of the Mohr-Coulomb failure criterion, where the yield stress is proportional to the ice overburden pressure, which in turn is proportional to the thickness of the ice column. In contrast, the profile of Ice Stream A is distinctly convex and shows a very low correlation between

the ice thickness and the driving stress. In this case, however, there is a quite strong correlation between the surface slope and the driving stress.





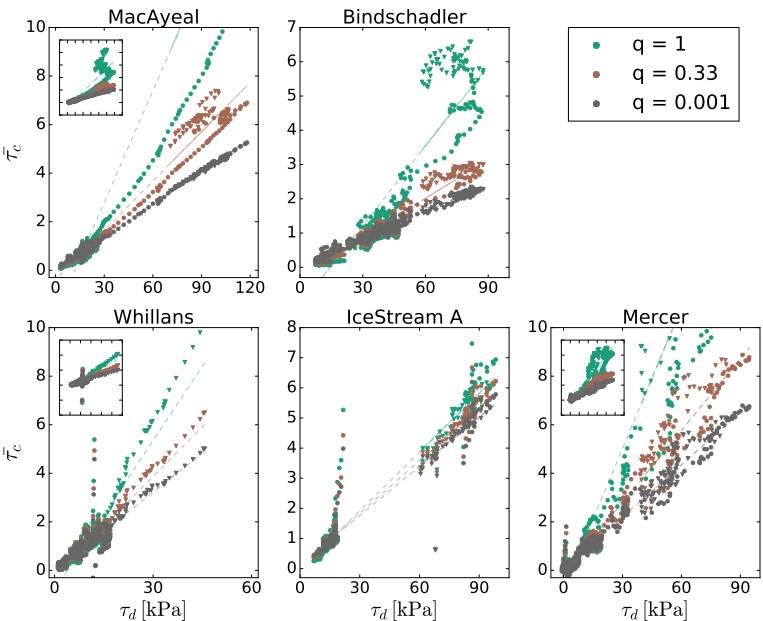

Figure 3: Normalized yield stresses $\bar{\tau}_c$ of Siple Coast ice streams as a function of the driving stress for different pseudo plastic exponents $q$. Normalization is computed by division with the median value. Triangles denote data on the uppermost 15% of the streams. In Whillans ice stream, negative values of the yield stress arise next to values where the membrane stress is undefined. Sometimes the main plot displays only parts of the data in order not to visually overstate effects at the stream onset or implausible negative yield stresses. In these cases, the entire data set is given in an inlay. Dashed lines are fitted to the data using a minimization of least squares method and a linear fit function.

| q=0.001, q=0.33, q=1 | Bindschadler | Mercer | IceStream A | Whillans | MacAyeal |
|---|---|---|---|---|---|
| driving stress | **0.98**, **0.96**, 0.84 | **0.97**, **0.96**, 0.89 | **0.92**, **0.92**, **0.91** | 0.79, 0.82, 0.84 | **1.0**, **0.97**, 0.84 |
| slope | 0.82, 0.71, 0.52 | 0.86, 0.8, 0.66 | 0.86, 0.85, 0.82 | 0.73, 0.73, 0.7 | **0.93**, 0.84, 0.63 |
| thickness | 0.84, 0.88, 0.83 | 0.7, 0.76, 0.84 | 0.28, 0.26, 0.23 | 0.65, 0.68, 0.67 | 0.79, 0.88, **0.93** |

Table 1: Correlation Coefficients $R^2$ between the yield stress and the driving stress, the slope or the thickness respectively. Three values are given in each table entry, corresponding to different choices of the pseudo plastic exponent $q$. Values greater or equal $0.9$ are put in bold font in order to highlight high correlations.




## 4 Conclusions

There are a number of assumptions that we have made in the analysis of the data and as a matter of course all results are subject to the constraints that arise through theses assumptions. For a one-
dimensional isothermal ice stream, where vertical shear in the ice column can be neglected, we showed that basal yield stress can be expressed in terms of driving stress and membrane stress and hence solely by along-flow geometrical properties, such as ice thickness, surface elevation and spatial derivatives thereof. Applied to five Siple Coast ice streams along their main flowlines, we find that the membrane stress is small in comparison to the driving stress. In fact, it can be neglected
over most of the stream length. Furthermore, we find very good correlation ($R^2 \geq 0.9$ in four of five cases) between the basal yield stress and the driving stress for comparably small values of the pseudo plastic exponent $q$ (Coulomb limit $q \to 0$ and the often in ice sheet modeling used value $q = 0.33$).

Since basal properties typically depend on a multitude of different properties and processes of the till and the hydrological system, which interact in complex ways, this linear relationship seems
surprisingly simple. We want to highlight that we do not suggest that the yield stress is independent from basal boundary conditions nor do we confuse correlation with causality. We rather hypothesize that in an ongoing interaction between ice dynamics and the hydrological and till system in the Siple Coast ice streams, a configuration is obtained where the yield stress follows the driving stress and vice versa.

Our simplified flowline analysis can not capture across-flow processes such as the convergence of different tributary flows, stream widening or the convergence of different streams (compare Whillans and Mercer ice stream in Fig. 1). The Shallow Shelf Approximation (SSA) may not be sufficient in the vicinity of the ice stream onset where vertical shear seems to be relevant. In addition, the used data set (Bedmap2) contains considerable uncertainty and noise, such that the data has been
subjected to a running-mean smoothing algorithm before the analysis. Nevertheless, regarding the high correlation, we consider our finding as significant in first approximation and a valid estimate for the yield stress underneath Siple Coast ice streams. Given that, for example, the Institute and Möller Ice Streams in the Weddell sector show comparable features to the Siple Coast Ice Streams in terms of a relatively flat surface profile and weak sediments (Siegert et al., 2016b), our results might be
applicable beyond the Siple Coast.

Future research questions could involve a process-based decomposition of the different contributors. Since an evolving ice softness as well as melting processes certainly play a major role in ice streams, the application of an energy model (non-isothermal case) and of an hydrological model should be elements of this investigation.





## Appendix: Expressing the membrane stress as a function of the ice topography

In Sec. 2 we derive an expression for the basal yield stress that depends only on the ice topography. We perform the calculation step by step in this section.

Following Eq. 4, the membrane stress is

$$\tau_m = 2(AE_{SSA})^{-\frac{1}{n}} \left[ H v_x^{\frac{1}{n}} \right]_x . \tag{12}$$

Exploiting the product and chain rule allows us to rewrite the non-trivial part of the above equation as

$$\left[ H v_x^{\frac{1}{n}} \right]_x = H_x (v_x)^{\frac{1}{n}} + \frac{H}{n} (v_x)^{\frac{1}{n}-1} v_{x^2}. \tag{13}$$

Based on Eq. 3, we can express the derivatives of the velocity $u$ as derivatives of the ice topography:

$$v_x = [(ax+Q_0)H^{-1}]_x = aH^{-1} - (ax+Q_0)H^{-2}H_x \tag{14}$$

and

$$v_{x^2} = \left[ aH^{-1} - (ax+Q_0)H^{-2}H_x \right]_x = -2aH^{-2}H_x - (ax+Q_0)\left\{ H^{-2}H_{x^2} - 2H^{-3}H_x^2 \right\}. \tag{15}$$

Inserting Eqs. 14 and 15 into Eq. 13 then yields

$$\left[ H v_x^{\frac{1}{n}} \right]_x = H_x [aH^{-1} - (ax+Q_0)H^{-2}H_x]^{\frac{1}{n}} + \frac{H}{n} (aH^{-1} - (ax+Q_0)H^{-2}H_x)^{\frac{1}{n}-1} \cdot$$
$$\cdot \left\{ -2aH^{-2}H_x - (ax+Q_0)\left\{ H^{-2}H_{x^2} - 2H^{-3}H_x^2 \right\} \right\}, \tag{16}$$

which translates into Eq. 8, if inserted into Eq. 12.

*Acknowledgements.* We want to thank C. Schoof for clarification via personal communication. This work was supported by the DFG through Grants SPP 1158 (TA).





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
