# Peer review of "Balance between driving stress and basal drag results in linearity between driving stress and basal yield stress in Antarctica's Siple Coast Ice Streams"

_The Cryosphere, 2016_

## Referee Comment (RC1) · Anonymous Referee #1 · 30 Sep 2016

The paper entitled "Balance between driving stress and basal drag results in linearity between driving stress and basal yield stress in Antarctica's Siple Coast Ice Streams" by J. Wohland, T. Albrecht, and A. Levermann presents an analysis of the relationship between driving stress and basal yield stress along five flow lines in the Siple Coast Region. The authors find that membrane stresses are negligible and that there is a linear dependence between driving stress and basal yield stress.

[Figure]

**1 General Comments**

The paper is well written, and is easy to follow. I was a bit surprised by the abstract because it seemed to be a bit contradictory to our current understanding of ice stream dynamics. While it is now broadly accepted that membrane stresses are negligible near the divides and in slow moving regions, it has been shown in different studies that membrane stresses are key in order to explain ice acceleration and thinning when the grounding line (or ice front) retreats (e.g., Fürst et al., 2013; Seroussi et al., 2014). If membrane stresses were negligible, grounding line retreat would have no impact on ice velocity several kilometers upstream (just like with the Shallow Ice Approximation), which is not what we observe (e.g., Howat et al., 2008; Rignot et al., 2014).

Joughin et al. 2002 looked at velocity changes over the Siple coast region and, using a simple approach, found that basal friction was about half the driving stress. So, I wondered why the conclusions of this study were so different. In fact, the authors of this paper mention that they are using "inverse modeling" to infer basal stress, but if I understand correctly, InSAR derived velocities are not used here, except at the upstream boundary. The velocity is calculated using a simple, one dimensional equation that only relies on snow accumulation and the distance along the flow line:

$$u = \frac{Q_0 + ax}{H} \tag{1}$$

where $Q_0$ is the input flux, $a$ is the surface mass balance (SMB), $x$ is the coordinate along the flow line and $H$ the ice thickness. First, saying that $a = 0.5$ m/a is chosen because it is similar to MISMIP experiments is not a very convincing argument. Several model outputs from regional climate models are available online (MAR, RACMO) and show that the SMB in this region is more around 0.1 m/a. But, as the authors mention, this does not make a big difference in the calculation because $Q_0$ is much larger than $ax$, so we can assume that the flux is almost constant.

Now, what I am more worried about is that the authors use this very simple description of ice velocity to determine longitudinal drag and basal drag, instead of measured ice velocities. The problem with this approach is that the conclusions of the papers are entirely dependent on ice velocity since the ice velocity determines the stress regime. So I tried to do the same experiment and compared the velocity predicted by this simple model to measured surface velocities from Rignot et al. 2011. Figure 1 shows my 4 flow lines and figure 2 shows the difference between the "modeled" velocities (according to equation 1) and InSAR derived velocities along these flow lines, and the misfit is very large.

This difference is probably explained by the hypothesis of 1-dimensional flow, where ice coming from the sides is ignored. That is my second criticism. I don't believe that lateral drag can be neglected in these ice streams. Flow line models are good for process studies, but they are too simplistic to investigate real-life scenarios, as too many important factors are missing (lateral drag, and how it changes through time, flow convergence divergence, bump shapes, etc).

**2  Minor comments**

- l.10: the driving stress is linear in the basal yield stress → there is a linear relationship between the driving stress and the basal yield stress

- l.10: the ice "surface" topography ...

- l.14: remove "grounded" (ice sheets are grounded by definition)

- l.23: paradoxon → paradox

- l.36: summary

- l.73: membrane stresses include both longitudinal stress and lateral drag. You are assuming that lateral drag is negligible here.

- l.86: is represented here by the SSA

- l.146: include space between number and its unit

- l.227: the often used value in ice sheet modeling

- Conclusions: model limitations should be at the end of the discussion rather than in the conclusions

**3 References**

Fürst, J. J., H. Goelzer, and P. Huybrechts, Effect of higher-order stress gradients on the centennial mass evolution of the Greenland ice sheet, Cryosphere, 7(1), 183–199, doi:10.5194/tc-7-183-2013, 2013.

Howat, I. M., I. Joughin, M. Fahnestock, B. E. Smith, and T. A. Scambos, Synchronous retreat and acceleration of southeast Greenland outlet glaciers 2000-06: ice dynamics and coupling to climate, J. Glaciol., 54(187), 646– 660, 2008.

Joughin, I., S. Tulaczyk, R. Bindschadler, and S. Price, Changes in west antarctic ice stream velocities: Observation and analysis, J. Geophys. Res., 107(B11), 1–22, doi:10.1029/2001JB001029, 2002.

Rignot, E., J. Mouginot, and B. Scheuchl, Ice Flow of the Antarctic Ice Sheet, Science, 333(6048), 1427–1430, doi:10.1126/science.1208336, 2011.

Rignot, E., J. Mouginot, M. Morlighem, H. Seroussi, and B. Scheuchl, Widespread, rapid grounding line retreat of Pine Island, Thwaites, Smith and Kohler glaciers,

West Antarctica from 1992 to 2011, Geophys. Res. Lett., 41(10), 3502–3509, doi:10.1002/2014GL060140, 2014.

Seroussi, H., M. Morlighem, E. Rignot, J. Mouginot, E. Larour, M. P. Schod- lok, and A. Khazendar, Sensitivity of the dynamics of Pine Island Glacier, West Antarctica, to climate forcing for the next 50 years, Cryosphere, 8 (5), 1699–1710, doi:10.5194/tc-8-1699-2014, 2014.
* * *
[Figure]

**Fig. 1.** Flow line locations

[Figure]

[Figure]

**Fig. 2.** Calculated vs InSAR derived velocities

---

## Short Comment (SC1) · 20 Oct 2016

We thank the first reviewer for his/her constructive and helpful comments. We will provide a detailed response later, but would like to take the chance for a short comment on one important issue the reviewer pointed out: referring to Fig. 2 of the reviewer's comment we tried to reproduce the comparison of measured InSAR derived flow line velocities (Rignot et al. 2011) with those inferred from accumulation, ice thickness and inflow at the upper bound.

For the data we used (Bedmap2) and constant accumulation a=0.5m/a we obtain a

[Figure]

considerable smaller misfit between the two (same order of magnitude, see Fig. 1). For instance, the main trend is well reproduced in Bindschadler and Ice Sream A. As far as Mercer and Whillans are concerned, our analysis underestimates surface velocities at about $x=250$km and overestimates them closer to the grounding line. These deviations coincide well with areas of stream narrowing and widening (compare Fig. 1 in the discussion paper) and might thus be explained by geometrical properties. The remaining misfit could be explained by important side factors such as lateral drag, as the reviewer suggested. We will elaborate on this factors in our detailed response soon.

[Figure]

**Fig. 1.** Calculated vs. InSAR derived velocities for a=0.5m/a, compare Fig. 2 of first reviewer's response

---

## Referee Comment (RC2) · Anonymous Referee #2 · 18 Nov 2016

Review of "Balance between driving stress and basal drag results in linearity between driving stress and basal yield stress in Antarctica's Siple Coast Ice Streams"
By: J. Wohland et al.
Submitted to TCD on 12 September 2016

The authors investigate the relationship between ice stream geometry and basal stress. The primary contribution of this work is a simplified flowline model, which the authors use with observed flow velocities and ice stream geometry to infer stresses at the bed and along the flowline within the ice column (sometimes called 'longitudinal stress' and herein called 'membrane stress'). The results indicate, as the title suggests, that membrane stresses are negligible and, consequently, that driving stress is a linear function of basal yield stress.

I like several things about the paper but have some major concerns. These concerns would need to be addressed before I would recommend the paper for publication.

Starting with the positives:
- Overall the manuscript is very clear and well written.
- The treatment of the model and underlying assumptions is concise and well organized, although it misses the most consequential assumption (see my main concern below). In particular, the authors go out of their way to qualify their results in light of the assumptions that they mention.
- The authors do a worthwhile error analysis for driving stress based on the errors available for the observations.
- The figures are neat and do a good job of communicating the main findings.

My main concerns are that the simplified model the authors use does not seem valid for this study and that the main conclusion drawn from this model are at odds with previous studies that the authors do not cite. In deriving the model, the authors neglect shearing in the margins in the SSA momentum equation (Eq. 1) without justification. Shearing in the margins can be a major component of the stress balance, particularly in Siple Coast ice streams. For example, Echelmeyer et al. (1994) found that shearing in the margins balances half of the driving stress in parts of Whillans Ice Stream while basal shear stress accounts for the other half of driving stress. There are flowline models that parameterize shear stress in the margins (e.g. Dupont and Alley 2006) and these seem more appropriate to this study than the model that is used. At the very least, the authors need to do more to convince the reader that their main conclusion makes sense in light of Echelmeyer et al. (1994), Tulaczyk *et al.* (2000a,b), and other studies on Siple Coast ice streams that point to drag in the margins as being a major factor in the stress balance.

Other major comments:

1. There is a lot of literature on basal shear stress and basal conditions in Siple Coast ice streams that is relevant to this study and that the authors do not mention. Certainly Tulaczyk *et al.* 2000a,b warrants some discussion. Many others have used ice geometry and flow velocities to constrain ice flow models and to infer basal conditions in the Siple Coast. This work should be taken into account. Examples include classic papers like MacAyeal (1989,1992) and Joughin *et al.* (2004), as well as more recent work by Morlighem *et al.* (2013) and Arthern *et al.* (2015).

2. Even though longitudinal stresses are unlikely to be important, smoothing the surface topography and ice thickness over 50 km (i.e. roughly 50 ice thicknesses) seems extreme

(see lines 144-155). Given the sensitivity of the results to ice geometry, this choice warrants a more thorough evaluation than is currently given. Kamb and Echelmeyer (1986a) provide a detailed, physically based discussion of how filtering can influence ice flow calculations. They argue that a rectangular window (like that used by the authors here) is a "poor approximation," preferring a triangular filter window. The length scale used by the authors here exceeds the length scales recommended by Kamb and Echelmeyer by a factor of 5 or more. Furthermore, other approaches that are more sophisticated and almost certainly better than the low-pass filtering applied by the authors are available to extract realistic gradients from noisy data (e.g. polynomial fitting, spline fitting, Savitzky-Golay filtering, etc.). Any of these approaches seem more appropriate for this work and all are available in most scientific programming packages (Matlab, Scipy, etc.), making them relatively straightforward to implement. At the very least, the authors should show how sensitive their conclusions are to their choice of smoothing window size.

Grammatical and minor comments:

Line 36: 'summary' is misspelled

Line 41: 'inverse' should be 'invert'

Line 62: 'hint to' should be 'hint at'

Line 82: 'theses' should be 'these'

Eq. 4: $v_x$ should be $u_x$

Line 120: The assumption of spatially constant A is discussed earlier (Line 91) and does not necessarily need to be repeated here.

Figures 1b and 2: On my printer, it is difficult to tell the difference between some of the line colors. I suggest using a different color scheme for the lines and/or using different line types or markers to distinguish between the different ice streams.